# Successful Treatment of Captive Common Marmosets (*Callithrix jacchus*) Infested with Common Cat Fleas (C*tenocephalides felis*) by Using Topical Imidacloprid and Environmental Control Measures

**DOI:** 10.3390/vetsci10090580

**Published:** 2023-09-18

**Authors:** Alexia Cermolacce, Romain Lacoste, Valérie Moulin, Amaury Briand, Jaco Bakker

**Affiliations:** 1Station of Primatology Centre National de la Recherche Scientifique (CNRS) UAR846, Route des Tours, 13790 Rousset, France; 2Department of Dermatology, Ecole Nationale Vétérinaire d’Alfort (ENVA), 94700 Maisons-Alfort, France; amaury.briand@vet-alfort.fr; 3Biomedical Primate Research Centre (BPRC), Animal Science Department (ASD), 2288 GJ Rijswijk, The Netherlands; bakker@bprc.nl

**Keywords:** flea, marmoset, imidacloprid, ectoparasite, nonhuman primate, New World monkey, pruritus

## Abstract

**Simple Summary:**

Fleas are one of the most frequently reported ectoparasites affecting birds and mammals. However, reports on captive nonhuman primates are extremely rare and lack details. This case report describes, in detail, the first case of a natural flea infestation by *Ctenocephalides felis* in a captive colony of common marmosets. The successful treatment of all the animals was achieved by using a combination of repeated topical administration of imidacloprid three weeks apart and decontamination of the animal enclosures. Reinfestation was prevented by stopping stray cats from entering the laundry rooms.

**Abstract:**

Fleas are ectoparasites affecting many animal species but reports in captive nonhuman primates are rare and mainly concern pet monkeys. Moreover, to the authors’ knowledge, a detailed report on marmosets is not known at present. This case describes the clinical signs, diagnosis, treatment and follow-up of a flea infestation by *Ctenocephalides felis* in a captive colony of common marmosets. Fleas, flea feces and skin lesions were identified on two animals during annual health screening. Subsequently, the entire colony was examined, and nearly half of the colony showed signs of infestation. Consequently, treatment was initiated for the entire colony and the environment. Animals received two topical administrations of imidacloprid (5 mg for animals <200 g and 10 mg for animals weighing >200 g) three weeks apart, and their enclosures were decontaminated using vaporizers containing permethrin, piperonyl butoxide, and pyriproxyfen. Subsequently, skin lesions were resolved and no evidence of fleas were noticed. No side effects of the treatment were observed. Stray cats were identified as the source of the infestation. Their access to the animal-related rooms was stopped. No reinfestation has been reported for 3 years. The topical application of imidacloprid appeared effective with no adverse events occurring, so may be appropriate for use in other non-human primates.

## 1. Introduction

The common marmoset (*Callithrix jacchus*) is widely used in neuroscience and regenerative medicine research and is one of the most found callithrichid species kept in captivity [1]. Captive nonhuman primates (NHP) include those in zoos, laboratory animal facilities, sanctuaries, rehabilitation centers, and private collections of pets. Nonhuman primates can be infected with a variety of parasites, many of which could be considered zoonotic. Fleas are natural ectoparasites of many mammals and birds [2]. Flea infestations are mostly known as infesting households with cats or dogs [3,4,5]. Nonhuman primates are susceptible to flea infestations transmitted by other animal species and humans, but NHPs are not the primary hosts. However, there is a paucity of information regarding the clinical signs and treatment of flea infestations in NHP [6,7,8,9]. The successful treatment using topical medications as used in cats has been reported in NHP; however, there is a lack of details [8].

Here we describe the first detailed case of clinical signs, diagnosis, treatment, and follow-up of a flea infestation by *Ctenocephalides felis* in a captive colony of common marmosets. The use of imidacloprid and environmental control measures resulting in the resolution of the clinical signs are presented.

## 2. Case Description

“Station de primatologie” (Rousset, France) houses an outbred breeding colony of common marmosets. The colony was formed around 2018 and consisted initially of captive-bred marmosets obtained from various accredited suppliers. Later, new breeding lines were introduced on several occasions to maintain the outbred character of the colony. The colony was housed indoors, in three distinct rooms, with no outdoor access and only marmosets were present in the animal facility. Animals were housed, in pairs or families, in cages measuring (150 × 190 × 230 cm) with a solid bottom and contact bedding (wood chips). The bedding was maintained weekly by removing the soiled woodchips and supplementing it with fresh litter. Cages were cleaned with water and a detergent (VEGE 15^®^, Société Industrielle de Diffusion, Créteil, France) once every two to three months and the surrounding corridors were cleaned once weekly with the same detergent. The cages were enriched with branches and toys. The animal rooms were maintained at a temperature of 25–27 °C with a relative humidity of 40–60% and a 12 h light–dark cycle. The room ventilation rate was six air changes per hour. The daily diet consisted of commercial monkey pellets, limited amounts of fresh fruits and vegetables, and a homemade porridge supplemented with honey and vitamins A, C, D, and E. The animals had ad libitum access to tap water.

Qualified animal caretakers observed all animals for injuries, illness, and fecal consistency at least twice daily. Abnormalities were reported to the veterinarians, and daily health records were kept for each animal.

Every year, each marmoset underwent a complete physical examination including blood sampling to evaluate hematological and biochemical values. A tuberculin skin test was given to monitor colony tuberculosis status. In addition, the marmosets were monitored for the presence of various bacteria and parasites by examination of rectal swabs and fresh stool samples. The standard health program included the subcutaneous administration of a broad-spectrum antiparasitic drug ivermectin at the dose of 0.2 mg/kg.

At the time of this case, the colony was composed of 40 animals, divided into 10 groups. During the physical examinations in December 2019, two 2-year-old sibling male marmosets showed skin lesions, with apparent living fleas and flea feces in the fur (Figure 1 and Figure 2). The skin lesions were mainly located on the abdomen, especially in inguinal regions with thickening of the skin, cutaneous erythema, papules, and crusted plaques. The lesions appeared pruritic in both individuals. In addition, they showed spiky hair at the tail base.

Fleas were collected and observed under stereomicroscope (Nikon, Tokyo, Japan) (Figure 3). The species was diagnosed as *Ctenocephalides felis*.

Fleas are known to be contagious. To assess the extent of the infestation, the entire colony was physically examined. In the same room as the infested pair, two other families were housed. In a family consisting of six animals, flea feces were found on four animals and living fleas on two animals. In the other family, consisting of eight animals, four animals presented crusty lesions and papules but no living fleas; one animal presented skin lesions, flea feces, and living fleas; one animal presented living fleas; one animal showed flea feces only; and one animal seemed to be non-infested. In the second room, hosting three groups, no lesions or parasites were observed on the fourteen marmosets. However, pruritus was witnessed on one animal. In the third room, ten marmosets were divided into four groups. In two cages, no abnormalities were observed (in a total of six animals). In the other two other cages skin lesions were noted on all animals (in a total of four animals).

In conclusion, the entire colony appeared to be infested or was suspected to be infested. Therefore, treatment of both environment and animals was initiated. All animals were captured and moved to kennels to empty the rooms. Wood chips were removed, all cages were pre-washed with tap water and subsequently sanitized using a 5% solution of the bactericidal, virucidal, and fungicidal disinfectant SPECTRAL^®^ (Société Industrielle de Diffusion, Créteil, France), applied for a minimum contact time of 5 min. This preparation is based on an alkaline complexing agent, isopropyl alcohol, and benzalkonium chloride. Afterwards, the cages were not rinsed as it was not necessary nor recommended by the manufacturer. In addition, the air ventilation was shut down and a vaporizer was used to decontaminate the rooms and cages with PARASTOP Plus^®^ (diffuseur 75 mL, Virbac, Carros, France) containing permethrin, piperonyl butoxide, and pyriproxyfen. As the manufacturers’ recommended time for application was a minimum of three hours after total diffusion of the vaporizer, the decision was made to leave it in the rooms the entire night. To avoid the potential toxicity of the applied environmental decontamination, the animals spent the night in kennels away from the rooms. Twenty-four hours after the application of the vaporizer the air ventilation was restarted and after one hour the animals returned to their home cages.

Meanwhile, each marmoset was treated topically on the neck skin using imidacloprid spot on (ADVANTAGE^®^ 40 spot on for cats and rabbits, Elanco, Sèvres, France). Imidacloprid belongs to the group of insecticidal neonicotinoids that are able to block nicotinic acetyl-choline receptors, which are described to be important excitatory neurotransmitter receptors in the brain with selective toxicity for insects over vertebrates [10]. Applied dose: 5 mg for animals <200 g bodyweight and 10 mg for animals weighing >200 g. This treatment was repeated three weeks later. No adverse events occurred.

Moreover, the source of the contamination was investigated. As direct contact with wild, synanthropic or domestic animals was not possible, an indirect source of contamination was suspected. All staff (animal keepers and veterinarians) changed their clothes on arrival at the institute: they wore dedicated work clothes and shoes. The laundry storage room was inspected, and flea dirt and fleas were observed on the stored clothes. This observation suggested that, although no flea dirt and fleas were initially observed on the clothes while they were being worn by the staff in the facility, fleas on the clothes/shoes were transferred via what was taken from the laundry and worn. Interestingly, the involved staff never showed any signs of a flea infestation. Stray cats had access to the laundry storage room. Most likely, infested stray cats disseminated the fleas, pupaes and/or flea eggs indirectly to the marmosets. Subsequently, the laundry storage room was decontaminated using a 5% solution of SPECTRAL^®^, Société Industrielle de Diffusion, France, all clothes were washed, and access for the cats was blocked permanently.

All animals were reexamined two months post-treatment and no fleas or flea feces were observed. Skin lesions regressed and disappeared in all individuals without applying any other treatment. Afterwards, all animals undergoing annual screening were closely examined for flea infestation and no new case has been reported for more than three years.

## 3. Discussion

Literature concerning flea infestations in NHP is lacking details. To the authors’ knowledge, this is the first case report detailing signs, diagnosis, and successful treatment of fleas in common marmosets.

*Ctenocephalides felis* is a common ectoparasite for animals like dogs and cats [2,3,4,5]. A survey performed on cats infested by fleas in France showed the presence of eight different species of fleas with *Ctenocephalides felis* the most prevalent (97.9%). Cat infestation was observed all over the country and on all animals regardless of living inside or outside [11]. In our case, the infested marmosets were housed indoors in closed and controlled animal facilities. The marmosets had no outdoor access or to other animals. However, the marmosets belong to a large institute with several buildings on site. The origin of contamination showed to be infested strayed cats, having access to the laundry storage room. The infested cats likely deposited fleas and eggs in the laundry and those eggs subsequently hatched. The staff disseminated the fleas to the animals by wearing infested clothes. As it was impossible to trap those cats, flea treatment was never initiated.

The institute also housed other species of NHP such as baboons (*Papio papio* and *Papio anubis*), squirrel monkeys (*Saimiri scieureus*), and macaques (*Macacca mulatta*) but none of these species were infested with fleas. Therefore, treatment was never indicated. Marmosets were housed in cages with wood chips as bedding (deep litter) which was only removed for cleaning every two to three months. In contrast, the baboons and macaques were housed in cages with tile flooring, which were daily cleaned with water. The squirrel monkeys were housed in cages with wood chips as bedding, which was weekly removed (and substituted) for cleaning. This difference in cleaning procedure-providing bedding substrate might explain that fleas could survive and multiply in the marmoset facility.

Imidacloprid is a registered topical treatment for flea infestation in dogs and cats [3,4,12,13]. Many commercial presentations are available for pet owners, with the active substance alone or in combination with other anti-parasitic drugs. Johnson-Delaney reported successful treatment of flea infestation in NHP using imidacloprid at ‘feline doses’ (no more details provided) [8], that is why imidacloprid was elected in this case. Effects of imidacloprid are described to be remanent for three to four weeks in cats. In case of severe infestation, it is recommended to repeat the treatment after three weeks [3,4]. According to these recommendations, doses were adapted to marmoset weight and two applications were performed three weeks apart. Treatment was successful and no side effects were observed. As cat flea strains are reported to become resistant to permethrin, an alternative intervention is the use of spray IGR pyriproxyfen on the bedding materials and treatment of the marmosets with imidacloprid [14].

Flea infestations spread through eggs and adult fleas [15,16,17,18,19]. However, adult fleas typically do not leave a preferred host. Marmosets are probably not a highly preferred host as the inspections highlighted small numbers of adult fleas. If they were, there would have been hundreds of fleas in just a few weeks. The control strategy of flea infestation in pets was based on the tripartite approach of treating the pet, the indoor environment and outdoors. But the availability of modern therapies, such as imidacloprid, to break the cat flea life cycle and prevent reproduction has allowed for the stand-alone treatments that are applied to the pets. Therefore, indoor environment treatment is no longer recommended [20]. However, maintaining a clean as possible environment for pets is still recommended, e.g., rooms with carpets should be vacuumed regularly. The wood chips in the marmoset enclosures provide a good environment for flea development and reproduction. However, the disadvantages of using deep litter do not outweigh its advantages. The potential negative aspects of providing deep litter as bedding on animal wellbeings include possible excessive vermin reproduction and increased ammonia and moisture levels. In contrast, providing deep litter creates enrichment for the animals, a welfare benefit. Furthermore, the decrease in the frequency with which cages need to be cleaned means less frequent handling and disturbance of the animals. Moreover, regular cleaning of marmoset facilities cannot be performed as scent-marking is an important behavior in this species (olfactory communication) [21,22]. Regular cleaning can cause some degree of social disruption which could affect animal welfare negatively. Therefore, additional indoor environmental treatment using insecticide was only performed simultaneously with the first topical treatment.

## 4. Conclusions

To our knowledge, this is the first case report of flea infestation in captive common marmosets detailing clinical signs, diagnosis, and successful treatment. Topical treatment using imidacloprid combined with environmental decontamination enabled elimination of the ectoparasites. Clinical signs resolved without any other treatment. No side effects were observed. This treatment can probably be used in other NHPs. Identification of the source of contamination allowed the establishment of preventing actions which seemed sufficient as no reinfestation has been diagnosed over the past three years. All involved staff should constantly and vigilantly monitor their NHP for ectoparasites.

## Figures and Tables

**Figure 1 vetsci-10-00580-f001:**
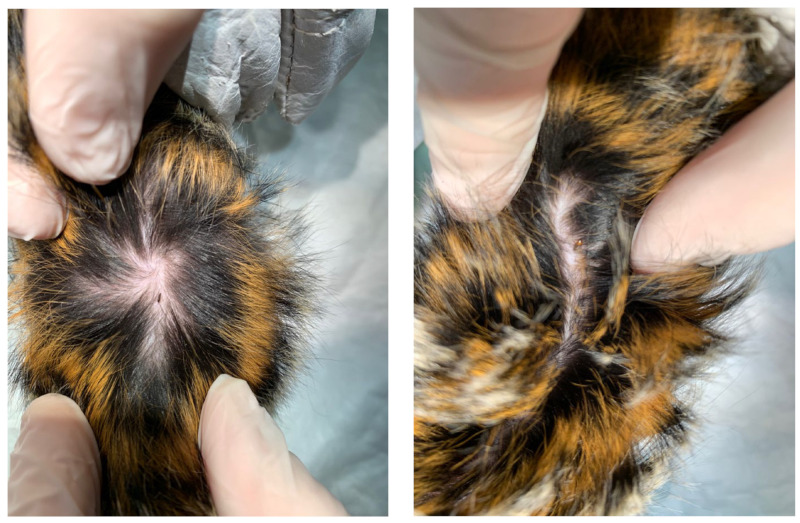
Living flea observed on the skin and in the fur of a marmoset (photographs provided by Alexia Cermolacce).

**Figure 2 vetsci-10-00580-f002:**
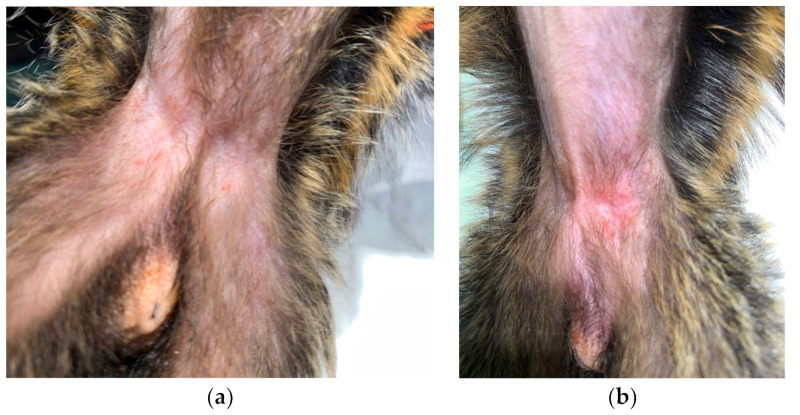
Observed skin lesions in the inguinal region of infested marmosets: papules (**a**), thickening of the skin and crusted plaque (**b**) (photographs provided by Alexia Cermolacce).

**Figure 3 vetsci-10-00580-f003:**
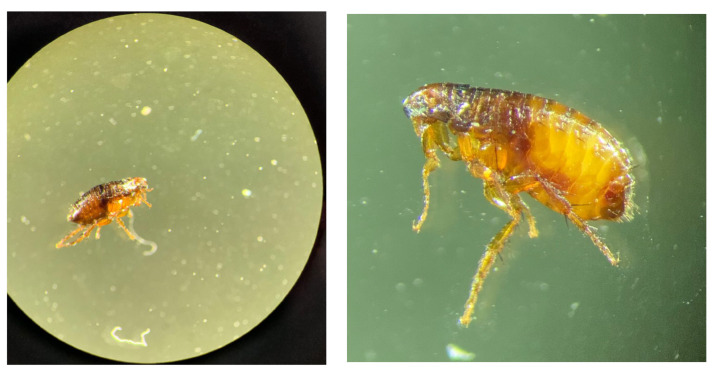
*Ctenocephalides felis* under stereomicroscope (photographs provided by Alexia Cermolacce). For identification, morphological characteristics were used: laterally compressed body shape, globulous head with perpendicular ctenidies and short tibias.

## Data Availability

Data are available on request.

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
