# Peer review of "Successful Treatment of Captive Common Marmosets (Callithrix jacchus) Infested with Common Cat Fleas (Ctenocephalides felis) by Using Topical Imidacloprid and Environmental Control Measures"

_vetsci, 2023, doi:10.3390/vetsci10090580_

Round 1

Reviewer 1 Report

1 The spectrum of imidacloprid for the treatment of parasites should be presented in the manuscript.

2 The use of disinfectants ((SPECTRAL®) and insecticidal compounds (SPECTRAL®) in the manuscript is not clear, and information on the composition of the relevant formulations of the products should be attached.

3 Case reports from dogs and cats in the area around the houses reflect the epidemiology of Ctenocephalides felis. But this part is missing from the manuscript.

4 Lack of environmental medication information. Lack of information on the use of imidacloprid in other animals at the centre.

Author Response

Comments to the reviewer 1

Dear reviewer 1,

We would like to thank you for your comments and remarks. We have revised the manuscript according to your remarks. Please find below our point-to-point reply:

Comment #1: The spectrum of imidacloprid for the treatment of parasites should be presented in the manuscript.

Response: The description of imidacloprid spectrum has been added at lines 187-208.

Comment #2: The use of disinfectants ((SPECTRAL®) and insecticidal compounds (SPECTRAL®) in the manuscript is not clear, and information on the composition of the relevant formulations of the products should be attached.

Response: The missing information has been added in lines 170-185.

Comment #3: Case reports from dogs and cats in the area around the houses reflect the epidemiology of Ctenocephalides felis. But this part is missing from the manuscript.

Response: We agree, therefore epidemic information of fleas in France is added in the discussion at line 234-236 including the reference.

Comment #4: Lack of environmental medication information. Lack of information on the use of imidacloprid in other animals at the centre.

Response: As requested this missing information is added in the discussion too (lines 245-247-222).

Kind regards,

Alexia

Author Response

Comments to the reviewer 2

Dear reviewer 2,

We would like to thank you for your comments and remarks. We have revised the manuscript according to your remarks. Please find below our point-to-point reply:

Comment of the reviewer It was not just imidacloprid – add to the title (and elsewhere as

appropriate) that environmental control measures were also used.

Answer of the authors: title and the rest of the manuscript has been adapted as suggested.

Comment of the reviewer As this is an English language journal, there is a need for English editing. This reviewer has suggested some edits, but many more are needed.
Answer of the authors: we agree, the whole manuscript has been edited.

Comment of the reviewer Although you state only marmosets are present in the facility, it would be helpful to be very clear in the paper that it is impossible for wildlife to enter.

Answer of the authors: line 211-212 should make this issue clear “As direct contact with other animals like wildlife, cats and/or dogs was not possible, an indirect source of contamination was suspected”.

Comment of the reviewer It is very interesting that the staff spread the infestation to the marmosets. I am struggling to understand how this occurs. Is it that the flea dirt on the clothes/shoes was transferred via what was taken from the laundry and worn. Suggest stating that no fleas were observed on the clothes while they were being worn when staff were in the facility with the marmosets. Suggest adding a statement that no staff showed any sign of a flea infestation.

Answer of the authors: we agree with the comment and edited line 214-219 to “The laundry storage room was inspected, and flea dirt and fleas were observed on the stored clothes. This observation suggests that, although no flea dirt and fleas were initially observed on the clothes while they were being worn by the staff in the facility, fleas on the clothes/shoes were transferred via what was taken from the laundry and worn. Moreover, the involved staff never showed any sign of a flea infestation.”

Comment of the reviewer 23, 50, 181 etc: Delete “symptoms” – A symptom is an effect noticed and experienced only by the person who has the condition, so cannot apply to animals.

Answer of the authors: deleted as requested.

Comment of the reviewer Line 19. Re-infection should be reinfestation.

Answer of the authors: corrected over the whole manuscript.

Comment of the reviewer Line 32: Add “the” before source; r

Answer of the authors: added as suggested.

Comment of the reviewer Line 34. This was not a safety study, so suggest to amend sentence to read “...effective with no adverse events occurring, so may be appropriate for use in other ...”

Answer of the authors: revised as suggested.

Comment of the reviewer General comment – please restructure sentences so they do not begin with an abbreviation (eg NHP), or spell out the abbreviation.

Answer of the authors: adapted over the whole manuscript.

Comment of the reviewer It was use (what is the difference between use and detailed use, so suggest deleting detailed) of imidacloprid, along with environmental control measures. Imidacloprid alone may not have been successful.

Answer of the authors: we agree with this remark and adapted this in the whole manuscript, including title.

Comment of the reviewer Line 60. Use same tense throughout – change are to were

Answer of the authors: in the whole manuscript the same tense is created.

Comment of the reviewer Stereo

Answer of the authors: a typo error occurred, adapted.

Comment of the reviewer Line 138. Delete a full stop.

Answer of the authors: this was an typo error, corrected.

Comment of the reviewer Fourteen
Answer of the authors: unfortunately again a typo error, corrected.

Comment of the reviewer Line 164. I believe you mean no side effects were observed

Answer of the authors: correct, changed in the manuscript to ‘no adverse events occurred’.

Comment of the reviewer Line 166. Indirect

Answer of the authors: really sorry but this was again a typo error, corrected.

Comment of the reviewer Line 167. Changed (not changes) their clothes on arrival at ...

Answer of the authors: see remark above. Same tense has been applied in the current manuscript.

Comment of the reviewer This is interesting. So there was no contact possible between the cats and the marmosets?

Answer of the authors: yes, correct, no direct contact was possible.

Comment of the reviewer Line 189. Surely you mean the cats deposited fleas and eggs in the laundry and those eggs subsequently hatched.

Answer of the authors: correct. Adapted as requested.

Comment of the reviewer Line 190. Why clean clothes if they were contaminated.

Answer of the authors: we agree that this is written confusing as the caretakers never noticed their clotes were infestated so the clothes were supposed to be clean. Adapted as requested.

Comment of the reviewer Line 212. Fleas spread through eggs is a misleading simplification. Please expand the description of the flea life cycle (see  Rust MK. The Biology and Ecology of Cat Fleas and Advancements in Their Pest Management: A Review. Insects. 2017 Oct 27;8(4):118; Rust MK, Dryden MW. The biology, ecology, and management of the cat flea. Annu Rev Entomol. 1997;42:451-73.).

Answer of the authors: correct. Adapted as requested and references are added.

Kind regards,

Alexia.

Reviewer 3 Report

General Comments. The article was interesting. It is still uncertain whether the cat fleas were successfully breeding after feeding on the marmosets and in the wood chips. The conditions seem ideal. However, the numbers of fleas within a month or so would have very high.

There are a few minor details that should be corrected.

Most of the cat flea strains that I have tested are resistant to permethrin. In the future, you might just spray the bedding materials with the IGR pyriproxyfen and treat the marmosets with the imidacloprid.

 Specific Comments:

 Lines 30 and 154. The fogger contains permethrin, piperonyl butoxide, and pyriproxyfen

Line 170 and 190. I think it is highly unlikely that pupae and flea eggs are being carried on the clothing. Eggs typically hatch within a few days and are susceptible to dessication. 

Line 212. Adult fleas do transfer from one host to another see Rust 1994 Interhost movement of adult cat fleas. J. Medical Entomology 31: 486-489. The adults typically do not leave a preferred host, but marmosets are probably not a highly preferred host. The inspections and the small numbers of adult fleas suggest marmosets are not preferred. If they were, there would have been hundreds of fleas in just a few weeks.

The article Franc et al. 2013. Direct transmission of cat flea between cats exhibiting social behavior. Parasite 20:49 could be added. It shows about 4% transmission during groomin and play. The authors must remember that cats are a highly preferred host. I would not consider marmosets as a preferred host and not expect the fleas to remain on them. 

Author Response

Comments to the reviewer 3

Dear reviewer 3,

We would like to thank you for your comments and remarks. We have revised the manuscript according to your remarks where possible. Please find below our point-to-point reply:

Comment of the reviewer General Comments. The article was interesting. It is still uncertain whether the cat fleas were successfully breeding after feeding on the marmosets and in the wood chips. The conditions seem ideal. However, the numbers of fleas within a month or so would have very high.

Answer of the authors: we agree and adapted lines 326-329 to ‘Flea infestations spread through eggs and adult fleas [15-19]. However, adult fleas typically do not leave a preferred host. Marmosets are probably not a highly preferred host as the inspections highlighted small numbers of adult fleas. If they were, there would have been hundreds of fleas in just a few weeks’.

Comment of the reviewer Most of the cat flea strains that I have tested are resistant to permethrin. In the future, you might just spray the bedding materials with the IGR pyriproxyfen and treat the marmosets with the imidacloprid.

Answer of the authors: we agree and added this in the discussion including a reference.

Comment of the reviewer Lines 30 and 154. The fogger contains permethrin, piperonyl butoxide, and pyriproxyfen.

Answer of the authors: correct, corrected in the whole manuscript.

Comment of the reviewer Line 170 and 190. I think it is highly unlikely that pupae and flea eggs are being carried on the clothing. Eggs typically hatch within a few days and are susceptible to dessication. 

Answer of the authors: corrected in the manuscript as suggested.

Comment of the reviewer Line 212. Adult fleas do transfer from one host to another see Rust 1994 Interhost movement of adult cat fleas. J. Medical Entomology 31: 486-489. The adults typically do not leave a preferred host, but marmosets are probably not a highly preferred host. The inspections and the small numbers of adult fleas suggest marmosets are not preferred. If they were, there would have been hundreds of fleas in just a few weeks.

The article Franc et al. 2013. Direct transmission of cat flea between cats exhibiting social behavior. Parasite 20:49 could be added. It shows about 4% transmission during groomin and play. The authors must remember that cats are a highly preferred host. I would not consider marmosets as a preferred host and not expect the fleas to remain on them. 

Answer of the authors: correct! Adapted in the manuscript including references.

Kind regards,

Alexia